# Field Modeling the Impact of Cracks on the Electroconductivity of Thin-Film Textronic Structures

**Stanisław Pawłowski** [1], **Jolanta Plewako** [2] and **Ewa Korzeniewska** [3,*]

1. Department of Electrodynamics and Electrical Machine Systems, Faculty of Electrical and Computer Engineering, Rzeszow University of Technology, 35-959 Rzeszow, Poland; spawlo@prz.edu.pl
2. Department of Power Electronics and Power Engineering, Faculty of Electrical and Computer Engineering, Rzeszow University of Technology, 35-959 Rzeszow, Poland; jplewako@prz.edu.pl
3. Institute of Electrical Engineering Systems, Faculty of Electrical Engineering, Electronics, Computer and Control Engineering, Lodz University of Technology, 90-924 Lodz, Poland
* Correspondence: ewa.korzeniewska@p.lodz.pl

**Abstract:** Wearable electronics are produced by depositing thin electroconductive layers with low resistance on flexible substrates. In the process of producing such metallic films, as well as during their usage, structural defects may appear which affect their electrical properties. In this paper, we present analytical and numerical models for understanding phenomena related to the electrical conductivity of thin electroconductive layers. The algorithm in the numerical model is based on the boundary integral equation method. The formulas enable calculation of the potential distribution and electric field strength of the analyzed structures, and describe the impact of cracks on their electrical resistance. The validity of the proposed models was verified by experimental results.

**Keywords:** thin films; wearable electronics; textronics; cracks; modeling of electroconductivity phenomena; electroconductivity; PVD

---

## 1. Introduction

Physical vapor deposition (PVD) is one of the best-known methods for producing thin metallic layers on various substrates. The PVD technique is a thin layer application process, whereby atom after atom is placed systematically and stochastically on the substrate, by the evaporation of the material from a heated source. Thin layers usually have a thickness ranging from atomic layers to several microns. The process changes both the surface properties and the transition zone between the substrate and the evaporated material. The common techniques for PVD of thin metallic layers are evaporation and atomization of droplets in a gaseous state. These techniques allow for precise deposition of particles on the substrate, while maintaining low pressure [1]. The adhesion of the deposited metallic particles depends on both the quality of the substrate and the size of the deposited particles. The larger the particle size, the lower the adhesion of the metallic layer to the substrate. The atomic deposition process is usually carried out in a vacuum or gas environment. The use of a vacuum ensures very low levels of gaseous pollutants. However, during deposition some impure particles may be released from the molten coating material and transferred to the substrate, reducing the purity of the obtained coatings. It is therefore important when using any substrate, including flexible substrates, to maintain the purity of the source material, in order to produce high-purity electrically conductive structures. Moreover, during the deposition of metal in a gaseous state on substrates with much lower temperatures there may be thermal stresses, which (as in the cases described in this paper) can lead to significant structural defects.

Defects in bulk materials differ from those in thin films. In bulk materials, damage is the result of sudden cracks. A single crack forms suddenly and propagates in a very short time, leading to material damage. Defects in the form of cracks in thin layers usually result from the relatively slow spread of many cracks. Electrical resistance increases with the number and size of the cracks [2].

In the process of depositing thin micron-sized films, areas are created covering several grains with a soft orientation. These can constitute significant geometrical imperfections in thin layers. Damage occurs as a result of the accumulation of defects arising primarily at the grain boundary, causing high stress levels in those areas, or due to the shifting of grain boundaries by processing defects, such as stretching or bending [3–6]. Electroconductive thin layers applied to elastic layers are subjected to constant bending and tensile mechanical stress. The resulting defects usually have the shape of thin cracks (Figure 1), which can have a very significant impact on the electroconductive properties of the layer. The resistance of thin layers is also affected by the anisotropic properties of the substrate [7]. The impact of defects on the electrical properties of films has been discussed by [8–15].

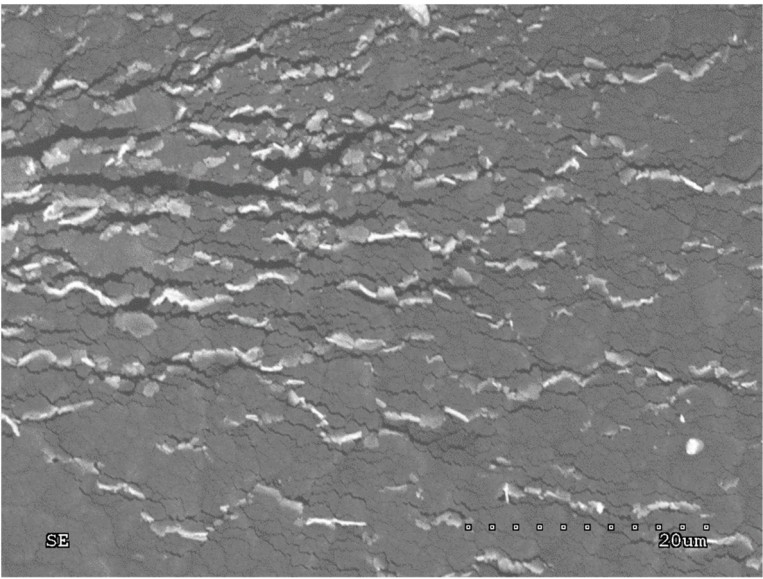

**Figure 1.** Microscopic image of the surface of samples with a thin Ag layer subjected to cyclic bending stress.

Physical vacuum deposition is used primarily when a high deposition index is required for a high-quality thin layer. Using PVD technology, it is possible to produce coatings with excellent quality adhesion, homogeneous layers, and designed structures, as well as layers with variable properties and controlled morphology [16–21]. This technology has many functional applications, such as for coating tools, decorative elements, optical enhancements, forms, matrixes, and blades, to name but a few of their current applications [22–25]. The authors of the present work are developing the PVD technique for use in the production of electrically conductive elements applied in textronics [26]. Textronic elements employ electrically conductive films produced on textile substrates for use in wearable electronics and small portable devices [27–29]. High demands are placed on such elements, not only in terms of mechanical resistance but also and more importantly in terms of electrical conductivity.

Despite many studies regarding thin layers themselves, there are few descriptions in the literature of physical phenomena occurring in thin-film structures produced on elastic dielectric substrates, such as textile composites. The purpose of this work was to create analytical and numerical models based on field theory to enable analysis of phenomena occurring in thin layers, and the impact of defects on the resistance of textronic electroconducting paths. Such understanding is particularly important when designing wearable electronics structures and predicting the behavior of electrically conductive layers during their usage.

## 2. Materials and Methods

### 2.1. PVD Deposition

The vacuum deposition process was carried out using a Pfeiffer Vaccuum Classic 250 chamber. It was conducted for 5 min, after obtaining an initial vacuum of $5 \times 10^{-5}$ mbar (0.005 Pa). Silver and gold with a purity of 99.99% guaranteed by Mennica Metale Ltd. Poland were used as the deposited material. The metals were evaporated from a tungsten boat with a higher melting point (3422 °C) compared to that of gold (2700 °C) or silver (2162 °C), which was used as a thermal source. The substrate was placed 6 cm from the evaporation source. In the process of vacuum deposition, the position of the substrate was changed in a plane parallel to the evaporation source, to ensure uniform metal deposition on the substrate. A Cordura textile composite (Figure 2) was used as the substrate. This composite is used in the production of textile products with high strength requirements. It is composed of nylon (polyamide) fibers in a 2/1 twill weave with a thin polyurethane film. The homogeneous surface layer of the composite guarantees the possibility of producing a continuous electrically conductive structure with good adhesion. The polyurethane layer also has the tasks of absorbing any mechanical stress to which the textronic structure is subjected, and removing the Joul heat generated as a result of resistance and electric current flow.

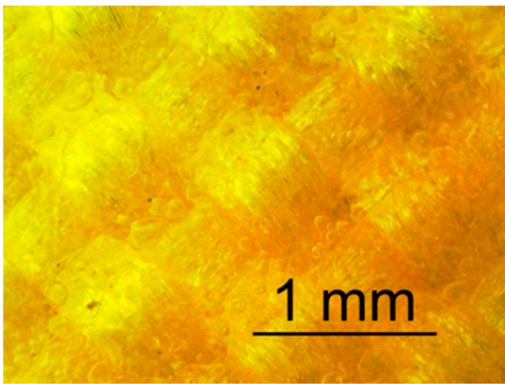

**Figure 2.** Microscopic image of the surface of the composite substrate on which the analyzed metallic layer was deposited (Optical SZ 630-T optical microscope with a magnification of 80 times).

### 2.2. Microscopic Research

Microscopic examination of the produced layers was carried out using a Hitachi S-4200 scanning electron microscope. The total width of the area in the SEM images was 600 μm.

## 3. Analytical Model

### 3.1. Model Assumptions

With some simplifying assumptions, a solution to the problem of a flow field in a thin conductive layer with an ellipse-shaped defect can be obtained analytically. The geometry of the model is shown in Figure 3.

The following assumptions were made:

1.  The medium of the area surrounding the defect is a homogeneous, isotropic, and linear conductor.
2.  The defect area is an ideal dielectric.
3.  The primary electric field $E_0$, forcing the current flow is constant and homogeneous (by primary field, it means those that would appear in the conductive area under consideration without a defect).
4.  Field functions do not depend on the coordinate directed perpendicular to the surface of the layer.

5.　The influence of the edges of the conductive area on the flow field distribution is neglected (i.e., it is assumed that the size of the defect is significantly smaller than the size of the conductive area and is located at a large distance from the edge).

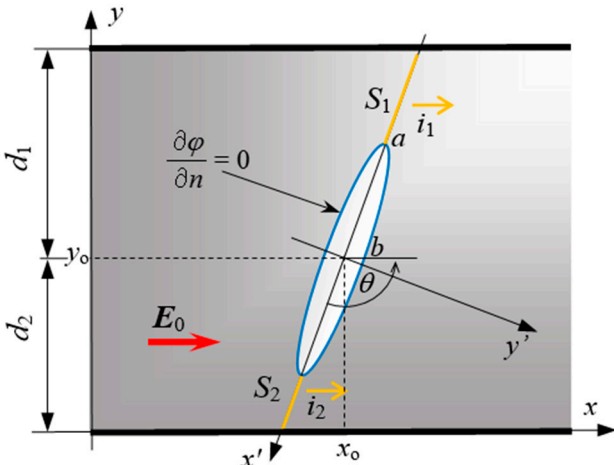

**Figure 3.** Analyzed model of a conductive layer with an elliptical defect.

### 3.2. Formulation of the Issue

Given these assumptions, in both the conductive and dielectric areas the electric field has scalar potential

$$\mathbf{E} = -\mathbf{grad}\phi \tag{1}$$

and fulfils the Laplace formula

$$\Delta\phi = 0 \tag{2}$$

At the boundary of the conductive and dielectric areas $\Gamma$, the potential function must be continuous and meet the zero Neumann condition from the side of the conductive area:

$$\left.\frac{\partial\phi}{\partial n}\right|_{\Gamma^+} = 0 \tag{3}$$

(the symbol $\Gamma^+$ indicates the boundary surface from the conductor side).

The sought distribution of the current density field is obtained by solving the problem formulated by Formulas (2), (3), and then using (1) and the local Ohm's law

$$\mathbf{J} = \gamma\mathbf{E} \tag{4}$$

### 3.3. The Solution in the Conductive Area

It results from assumption 5 that the formulated issue can be treated as an open issue, i.e., without an external edge. However, in what follows the authors wish to use the solution to estimate the current and resistance of the conductive path limited by the outer edges (more in Section 4). For this reason, in the analyzed model it is convenient to introduce two Cartesian coordinate systems: without the prim index, associated with the edges of the conductive path, and with the prim index, associated with elliptic defect axes (see Figure 3). The relationships between the coordinates of each system are expressed by the following formulas:

$$\begin{aligned} x' &= (x - x_O)\cos\theta - (y - y_O)\sin\theta \\ y' &= (x - x_O)\sin\theta + (y - y_O)\cos\theta \end{aligned} \tag{5}$$

where $x_O$, $y_O$ are the coordinates of the ellipse center and $\theta$ is the angle between the systems.

The potential function in the conductive area is shown in the form:

$$\phi^{\mathrm{I}} = \phi_0 + \phi_{\mathrm{i}} \tag{6}$$

where $\phi_0$ is the primary field potential and $\phi_{\iota}$ is the induced potential caused by the presence of a defect.

In the coordinates with a prime index, the potential of the primary field is expressed by the relationship

$$\phi_0(x', y') = -E_0(x' \cos\theta + y' \sin\theta) \tag{7}$$

In order to find the $\phi_{\mathrm{i}}$ potential, the elliptical coordinates $\eta$, $\psi$ defined by the relations (8) were used:

$$\begin{cases} x' = c\cosh\eta\cos\psi \\ y' = c\sinh\eta\sin\psi \end{cases} \tag{8}$$

On the edge $\Gamma$, the $\eta$ coordinate assumes a constant value $\eta_0$ which can be differently associated with the parameters of the ellipse $a$, $b$, $c$ ($c$—focal length of the ellipse, $c^2 = a^2 - b^2$, $a > b$, see Figure 3.):

$$\eta_0 = \operatorname{arcosh}\frac{a}{c} = \ln\frac{a+b}{c} = \frac{1}{2}\ln\frac{a+b}{a-b} \tag{9}$$

Based on Formulas (3)–(7), in the conductive area:

$$\left.\frac{\partial\phi_i}{\partial n}\right|_{\Gamma^+} = \left.\frac{\partial\phi_i}{\partial n}\right|_{\eta=\eta_0^+} = E_0 c(\cos\theta\sinh\eta_0\cos\psi + \sin\theta\cosh\eta_0\sin\psi) \tag{10}$$

This relationship Formula (10) is the boundary condition for the induced potential function. In elliptic coordinates, Formula (2) has the form:

$$\frac{\partial^2\phi_i}{\partial\eta^2} + \frac{\partial^2\phi_i}{\partial\psi^2} = 0 \tag{11}$$

The solution to the problem formulated by Formulas (10), (11) was obtained by applying the variable separation method. As a result:

$$\phi_i(\eta, \psi) = -E_0 c\sqrt{\frac{a+b}{a-b}}[b\cos\theta\cos\psi + a\sin\theta\sin\psi]e^{-\eta} \tag{12}$$

After taking into account Formulas (6), (7), (8) the full potential function in the conductive area is obtained in the form

$$\phi^{\mathrm{I}}(\eta, \psi) = -E_0\sqrt{\frac{a+b}{a-b}}[a\cosh\eta - b\sinh\eta]\cos(\psi - \theta) \tag{13}$$

Using Formula (1) and the gradient operator in elliptic coordinates (9), we obtain

$$E_\eta^{\mathrm{I}}(\eta, \psi) = \frac{E_0(a\sinh\eta - b\cosh\eta)}{(a-b)\sqrt{\cosh^2\eta - \cos^2\psi}}\cos(\psi - \theta) \tag{14}$$

$$E_\psi^{\mathrm{I}}(\eta, \psi) = \frac{-E_0(a\cosh\eta - b\sinh\eta)}{(a-b)\sqrt{\cosh^2\eta - \cos^2\psi}}\sin(\psi - \theta) \tag{15}$$

Then, transforming Formulas (13)–(15) into the Cartesian system $x'$, $y'$ we obtain

$$\phi^{\mathrm{I}}(x', y') = \frac{E_0}{a-b} \cdot [(\mathrm{sgn}(x')bP_1 - ax')\cos\theta - (\mathrm{sgn}(y')aP_2 - by')\sin\theta] \tag{16}$$

$$E_{x'}^{\mathrm{I}} = \frac{-E_0}{a-b} \cdot \left[\left(\mathrm{sgn}(x')\frac{Q_1+1}{2P_1}bx' - a\right)\cos\theta - \mathrm{sgn}(y')\frac{Q_1-1}{2P_2}ax'\sin\theta\right] \tag{17}$$

$$E_{y'}^{\mathrm{I}} = \frac{-E_0}{a-b} \cdot \left[\mathrm{sgn}(x')\frac{Q_2-1}{2P_1}by'\cos\theta - \left(\mathrm{sgn}(y')\frac{Q_2+1}{2P_2}ax' - b\right)\sin\theta\right] \tag{18}$$

where:

$$P_1 = \sqrt{\frac{x'^2 - y'^2 - c^2 + D}{2}}, P_2 = \sqrt{\frac{-x'^2 + y'^2 + c^2 + D}{2}} \tag{19}$$

$$Q_1 = \frac{x'^2 + y'^2 - c^2}{D}, Q_2 = \frac{x'^2 + y'^2 + c^2}{D} \tag{20}$$

$$D = \sqrt{\left(x'^2 + y'^2 - c^2\right)^2 + (2cy')^2} \tag{21}$$

*3.4. Solution in the Defect Area*

Substituting Formulas (9) into (13), the potential at the boundary of the conductive area is expressed as

$$\phi^{\mathrm{I}}(\eta_0, \psi) = -E_0(a+b)\cos(\psi - \theta) \tag{22}$$

As the potential function must be continuous, the relationship Formula (22) is also a boundary condition for the potential in the defect area. Using variable separation again, the solution in this area is as follows:

$$\phi(\eta, \psi) = -E_0(a+b)\left(\frac{c}{a}\cosh\eta\cos\psi\cos\theta + \frac{c}{b}\sinh\eta\sin\psi\sin\theta\right) \tag{23}$$

which after transformation into the Cartesian system gives:

$$\phi^{\mathrm{II}}(x', y') = -E_0(a+b) \cdot \left(\frac{x'}{a}\cos\theta + \frac{y'}{b}\sin\theta\right) \tag{24}$$

$$E_{x'}^{\mathrm{II}} = E_0\frac{a+b}{a}\cos\theta \tag{25}$$

$$E_{y'}^{\mathrm{II}} = E_0\frac{a+b}{b}\sin\theta \tag{26}$$

This means that the electric field in the dielectric area is a homogeneous field with the value

$$E^{\mathrm{II}} = E_0(a+b)\sqrt{\frac{\cos^2\theta}{a^2} + \frac{\sin^2\theta}{b^2}} \tag{27}$$

The condition of tangent continuity of the electric field strength component shows that this formula also determines the maximum value of the electric field at the surface $\Gamma$ from the conductive area side. Hence, having used Formula (4), it is possible to easily determine the maximum value of current density in the analyzed system.

The relationship determining the direction of the electric field vector in the defect area can be expressed as

$$\frac{E_{y'}^{\mathrm{II}}}{E_{x'}^{\mathrm{II}}} = \frac{b}{a}\mathrm{tg}\theta \tag{28}$$

To present the described field functions in the $x$, $y$, coordinate system associated with the conductive path, the relations in Formula (5) should be substituted for the coordinates $x'$, $y'$.

### 3.5. Estimation of the Impact of the Defect on the Current Value

The presented solution allows the impact of the defect on the total current flowing in the conductive path, and thus on its resistance, to be estimated. Using Formulas (4), (15) and general dependence:

$$i = \iint_S \boldsymbol{J} \cdot \mathbf{ds} \tag{29}$$

Formulas can be analytically derived for the current flowing through the cross-sections of the conductive path along the axis of the ellipse (see $S_1$, $S_2$ in Figure 3). After calculating Formula (29) it was obtained:

$$\frac{i_a}{i_0} = \frac{a\left(\sqrt{d_1^2 - c^2\cos^2\theta} + \sqrt{d_2^2 - c^2\cos^2\theta}\right) - bl}{(a-b)l} \tag{30}$$

$$\frac{i_b}{i_0} = \frac{al - b\left(\sqrt{d_1^2 + c^2\cos^2\theta} + \sqrt{d_2^2 + c^2\cos^2\theta}\right)}{(a-b)l} \tag{31}$$

where $i_a$ is the current flowing through the cross section along the long axis of the ellipse ($y' = 0$), $i_b$ is the current flowing through the cross section along the short axis of the ellipse ($x' = 0$), and $i_0$ is the current flowing in the path without any defects [30].

The principle of charge conservation states that the same direct current must flow through any cross-section of the conductive path. However, Formulas (30) and (31) will generally give different results, because the presented solution does not take into account the boundary conditions on the actual path edges (i.e., the normal component of the current density vector does not equal zero – see Assumption 5). In such case, the approximation error depends mainly on the size of the cross-sections of the path $S_1$, $S_2$—i.e., indirectly on the angle of inclination $\theta$ and the $a/b$ ratio of the ellipse. Our tests show that for better estimation, the formula from Formulas (30), (31) should be chosen that corresponds to the smaller total cross-section of the path.

To estimate the effect of a defect on the path resistance $R$, it is also necessary to estimate the potential difference between its ends on the basis of Formula (16), and then to use Formula (30) or (31) and Ohm's law. This procedure is simple, but the general relationship is expressed by a rather extended formula which need not be presented here. In the case of a defect of which the longer axis is perpendicular to the path axis ($\theta = \pi/2$),

$$\frac{R}{R_0} = \frac{H}{l}\frac{a\left(\sqrt{x_0^2 + c^2} + \sqrt{(l - x_0)^2 + c^2}\right) - bl}{a\left(\sqrt{y_0^2 - c^2} + \sqrt{(H - y_0)^2 + c^2}\right) - bH} \tag{32}$$

where $R_0$ is the resistance of the path without a defect, $H$ is the width of the path, and $l$ is the length of the path.

### 3.6. Infinitely Thin Defect

As presented in Figure 3, the actual cracks in conductive textronic layers are usually very thin, so it seems justified to model them with an infinitely thin defect, assuming the shorter half-axis of the

ellipse $b = 0$. Formulas describing the flow field distribution and the equivalent of Formula (32) with such a simplification are presented below:

$$\phi^{\mathrm{I}}(x, y) = -E_0 \cdot [x \cos \theta - \mathrm{sgn}(y) P_2 \sin \theta] \tag{33}$$

$$E_x^{\mathrm{I}} = E_0 \left( \cos \theta - \mathrm{sgn}(y) \frac{Q_1 - 1}{2P_2} x \sin \theta \right) \tag{34}$$

$$E_y^{\mathrm{I}} = -\mathrm{sgn}(y) E_0 \frac{Q_2 + 1}{2P_2} x \sin \theta \tag{35}$$

$$\frac{R}{R_0} = \frac{H}{l} \frac{\sqrt{x_0^2 + a^2} + \sqrt{(l - x_0)^2 + a^2}}{\sqrt{y_0^2 - a^2} + \sqrt{(H - y_0)^2 + a^2}} \tag{36}$$

Figures 4 and 5 show an example of the potential distribution and current density vector for this case (for $\theta = \pi/2$).

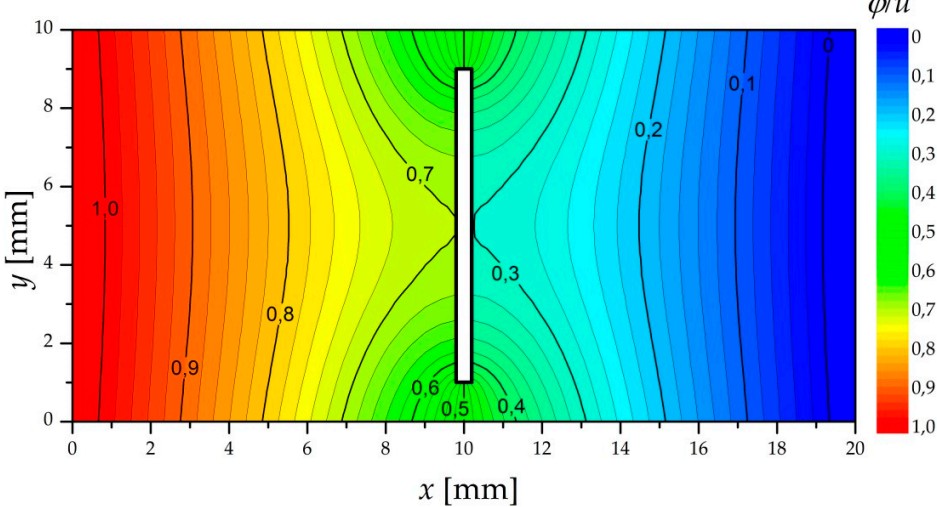

**Figure 4.** Potential distribution for an infinitely thin crack calculated on the basis of Formula (33) for $E_0 = 5$ V/m, $\theta = 90°$, $a = 8$ mm.

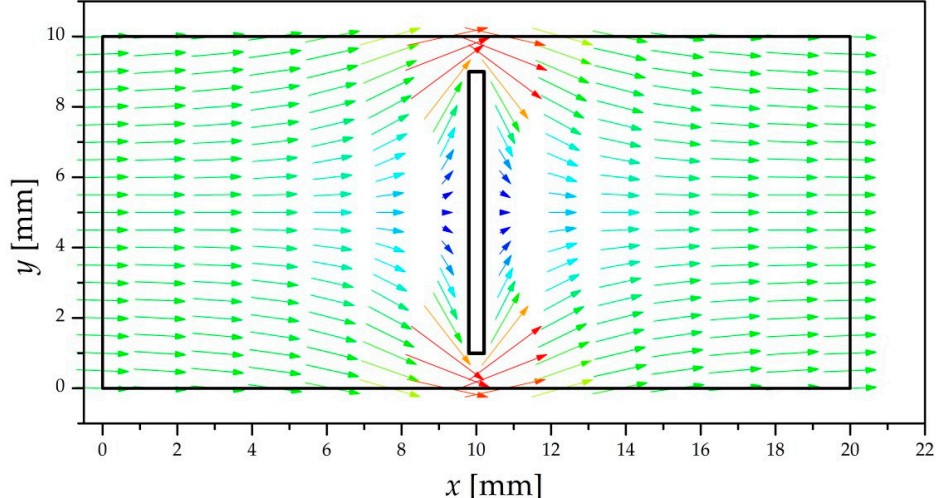

**Figure 5.** Distribution of the vector of current density for an infinitely thin crack calculated on the basis of Formula (33) for $E_0 = 5$ V/m, $\theta = 90°$, $a = 8$ mm.

Notice should be taken of the discontinuity in the potential function on the crack line, and the peculiarities of the electric field strength components at the ends of the defect segment.

## 4. Numerical Model

### 4.1. Formulation of the Issue

Solving the problem of boundary conditions at the edges of an actual conductive path with a defect requires the use of a numerical method. The geometry of the analyzed model is shown in Figure 6. For simplicity, it was initially assumed that the defect was infinitely thin and perpendicular to the path axis.

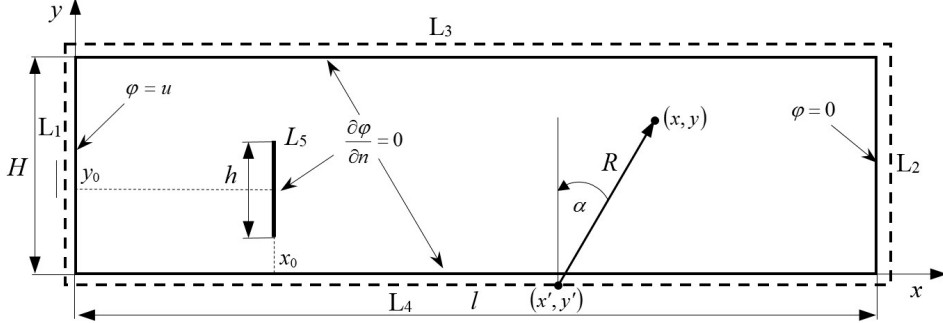

**Figure 6.** Analyzed model of conductive path with defect.

Boundary conditions at the path edges and the defect are described by dependencies

$$\phi(0, y) = u, \phi(l, y) = 0,$$
$$\left.\frac{\partial \phi}{\partial n}\right|_{y=0} = \left.\frac{\partial \phi}{\partial n}\right|_{y=H} = \frac{\partial \phi}{\partial n}|L_5 = 0 \tag{37}$$

The problem is to find a solution to Formula (2) with mixed boundary conditions Formula (37).

### 4.2. Solution Method

To solve the formulated problem, one of the commonly used numerical methods, such as the Finite Difference Method or the Finite Element Method can be used. However, the authors decided to use the less popular Integral Equation Method in this paper. It has several advantages over the previous ones. As the edge method, it does not require discretizing the entire analyzed area, only its edge (which significantly reduces the effective numerical model and speeds up calculations), the solution exactly meets the Laplace formula (the numerical error is connected only to meeting the boundary conditions), and also allows for a simple calculation of the potential derivatives which are needed to determine the current density distribution without the additional error.

To solve the problem as formulated, the boundary method of integral formulas is used. The searched potential function is presented in the form

$$\phi(x, y) = u\left(1 - \frac{x}{l}\right) + \sum_{k=1}^{5} \int_{L_k} \sigma_k(x', y') \frac{\cos \alpha}{R} dl_k \tag{38}$$

where: $\sigma_k(x', y')$ is the density potential functions of double layer on $L_k$ ($(x', y') \in L_k$) lines, $R$ is the distance between any point of the analyzed area and the current integration point (see Figure 6), $\alpha$ is the angle between the line connecting these points and normal to the line on the edge.

Part $u(1 - x/l)$ in (38) meets all the conditions of the problem except the boundary condition on the $L_5$ defect line. Thus, it corresponds to the original field. The second part representing the induced field is the sum of the double layer potentials. The density functions of these potentials are determined on

the $L_5$ defect line and the lines parallel to the path edges, which are pushed away a small distance from the analyzed area. This separation does not diminish the mathematical correctness and accuracy of the solution, but it avoids problems associated with singularities of underintegral functions and their discretization when calculating the integrals in Formula (37).

The use of double layer potentials is dictated by the fact that they are discontinuous functions on lines on which density functions are defined. This enables the step change in potential on an infinitely thin defect line to be taken into account (see Formula (33) and Figure 4). On lines $L_3$, $L_4$ the potential of a single layer could also be used (as in the Edge Element Method). However, it would be associated with a more burdensome numerical procedure.

The function class defined by Formula (38) meets Formula (2) exactly. The requirement for meeting boundary conditions leads to the following system of integral formulas:

$$\sum_{k=1}^{5} \int_{L_k} \sigma_k(x',y') \frac{\cos\alpha}{R} \, \mathrm{d}\, l_k = 0 \text{ for } x = 0 \text{ i } x = l \tag{39}$$

$$\frac{\partial}{\partial y} \sum_{k=1}^{5} \int_{L_k} \sigma_k(x',y') \frac{\cos\alpha}{R} \, \mathrm{d}\, l_k = 0 \text{ for } y = 0 \text{ i } y = H \tag{40}$$

$$\frac{\partial}{\partial y} \sum_{k=1}^{5} \int_{L_k} \sigma_k(x',y') \frac{\cos\alpha}{R} \, \mathrm{d}\, l_k = \frac{u}{l} \text{ for } x = x_0 \tag{41}$$

The following steps describe how to solve it:

1. Dividing each of the $L_k$ lines into $N_k$ sections (elements) and selecting $N_k$ collocation points on each of the shorelines.

2. Approximation of the potential density function on each element with constant functions, which allows the expression of integrals in Formulas (38)–(41) using elementary functions.

3. Substitution of coordinates of collocation points to (39), which brings the problem to the system of linear algebraic formulas for $\sigma_{k,n}$:

$$\sum_{k=1}^{5} A_{k,m,n} \sigma_{k,n} = B_m \tag{42}$$

where:

$$A_{k,m,n} = \frac{1}{2\pi} \left( \arctan\frac{s_m^* - s_n}{t_m^*} - \arctan\frac{s_m^* - s_{n-1}}{t_m^*} \right) \text{ for } k = 1, \, 2 \tag{43}$$

$$A_{k,m,n} = \frac{1}{2\pi} \left( \frac{1}{s_m^* - s_{n-1}} - \frac{1}{s_m^* - s_n} \right) \text{ for } k = 3, \, 4, 5 \tag{44}$$

$$B_{k,m} = 0 \text{ for } k = 1, \, 2, \, 3, \, 4 \tag{45}$$

$$B_{k,m} = \frac{u}{l} \text{ for } k = 5 \tag{46}$$

$s_m^*$, $t_m^*$—coordinates of the collocation point in the local system of the $m$-th element of discretization (see Figure 7)
$s_n$, $t_n$—local coordinates of the ends of the line elements $L_1$–$L_5$

4. Numerical solution of the system of Formulas (42) (e.g., by Gaussian elimination method).

5. Calculation of the potential function after substitution approximated potential density functions to Formula (38).

6. Calculation of the electric field strength based on Formula (1), current density based on Formula (4), and current value based on Formula (29).

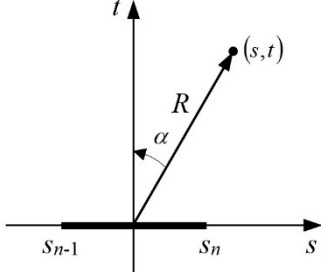

**Figure 7.** Element of discretization of lines on which the potential density function is determined (see dashed line in Figure 6) and the sense of local coordinates s, t.

### 4.3. Sample Results

The procedure described above was carried out in Microsoft Visual Studio 2010 in the Fortran 77 environment. Sample calculation results are presented in Figures 8 and 9. They correspond to analogous data used for the analytical calculations presented in Figures 4 and 5. The differences between these solutions can be seen primarily at the near the edge of the path. They result from the fact that the analytical model does not take into account the boundary conditions on these edges.

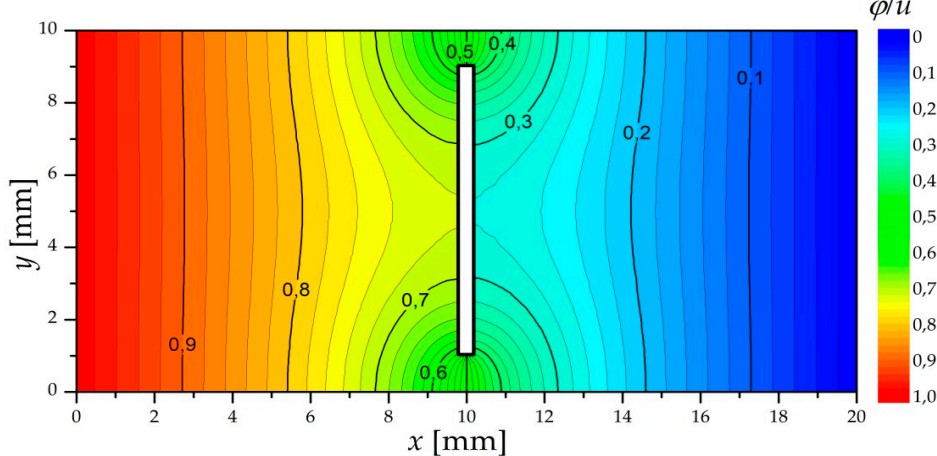

**Figure 8.** Potential distribution for an infinitely thin crack calculated using the numerical method for $u = 1$ V, $\theta = 90°$, $a = 8$ mm.

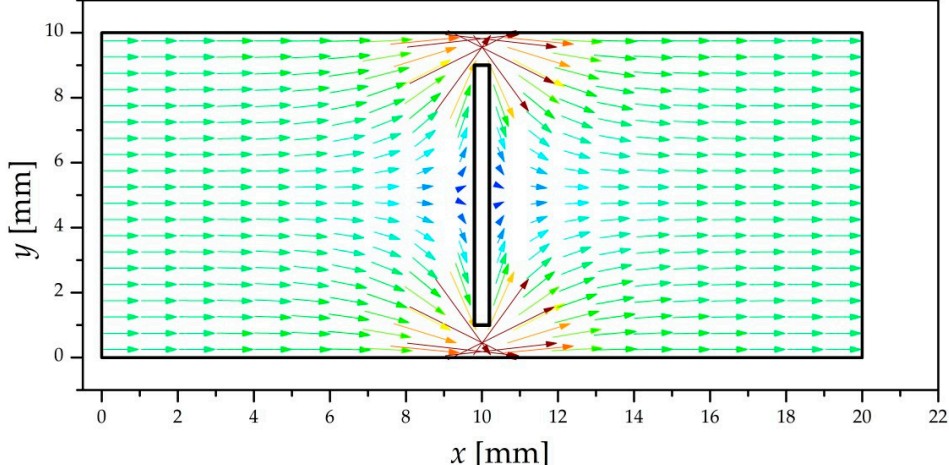

**Figure 9.** Distribution of the current density vector for an infinitely thin crack calculated by the numerical method for $u = 1$ V, $\theta = 90°$, $a = 8$ mm.

## 5. Measurement Verification

To verify the results of the analytical and numerical calculations, the physical model was submitted to a series of resistance measurements. Due to their small size and uneven surfaces, such measurements are quite troublesome on real textronic paths. For this reason, aluminum tape with a thickness 50 μm and width of 10 cm was used, with prepared incisions of various lengths (Figure 10). Two pairs of electrodes, current electrodes and voltage electrodes, were located at the ends of the tape. Their positions could be freely changed along the axis of the tape. The voltage electrodes provided equalization of potentials at the ends of the measured path segment, in accordance with the assumptions of the numerical model.

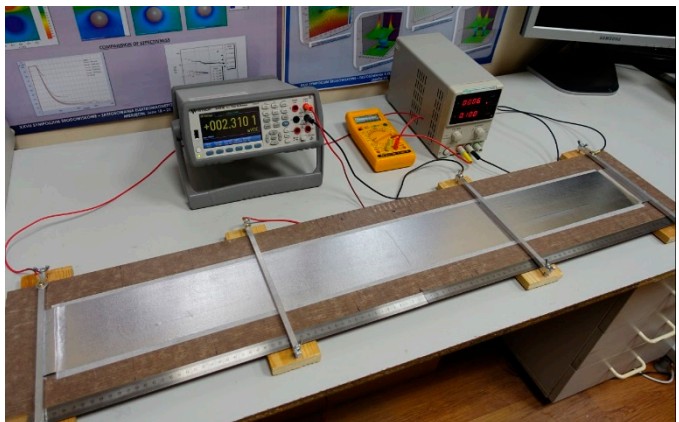

**Figure 10.** Resistance measurement system for a path with a defect (crack in the middle of the path).

To eliminate the influence of Joule's heat on the final results, all measurements were made at the same current value $i = 0.1$ A, obtained from a stabilized power supply. The cracks were made in such a way that their center coincided with the center of the tape, i.e., $x_0 = l/2$, $y_0 = H/2$ (Figure 5). The electric voltage between the voltage electrodes located perpendicular to the edge of the tape was measured. Different distances between the electrodes and different lengths of cuts were considered. The resistance was determined based on Ohm's law. Figure 11 presents the results of the measurements, combined with theoretical curves obtained from numerical simulations. Measurement uncertainties were estimated based on the accuracy of measuring instruments using the total differential method.

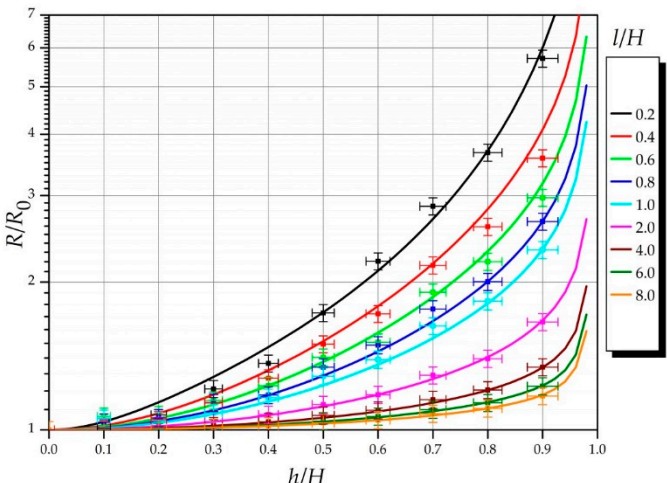

**Figure 11.** Comparison of experimental results and simulations of a defect located in the central part of the path.

In the vast majority of cases, the theoretical curves are within the limits of measurement errors. It can be concluded that the numerical simulations provide results consistent with measurements.

## 6. Estimating the Resistance of a Path with a Defect

The analytical solution presented in Section 3 is an exact solution for the analyzed model. However, the formula which is used for calculating the resistance Formula (36) and which is derived from the model should be treated as an approximate model. It does not take into account the real boundary conditions at the edges of the path. In the numerical model, these conditions are taken into account, so the solution should be considered more reliable. This is confirmed by comparison with the experimental tests described in the previous section. To assess its usability and scope of application, the values calculated from Formula (36) were compared with the results of calculations of path resistance made using the numerical model. The plot in Figure 12 illustrates the results for a defect located in the center of the path ($x_0 = l/2$, $y_0 = H/2$). It presents the relative percentage deviation between the resistance calculated using Formula (36) and the resistance $\widetilde{R}$ calculated numerically:

$$\delta R_\% = \frac{\left|\widetilde{R} - R\right|}{R} \cdot 100 \tag{47}$$

depending on the width of the defect and the length of the path related to its width.

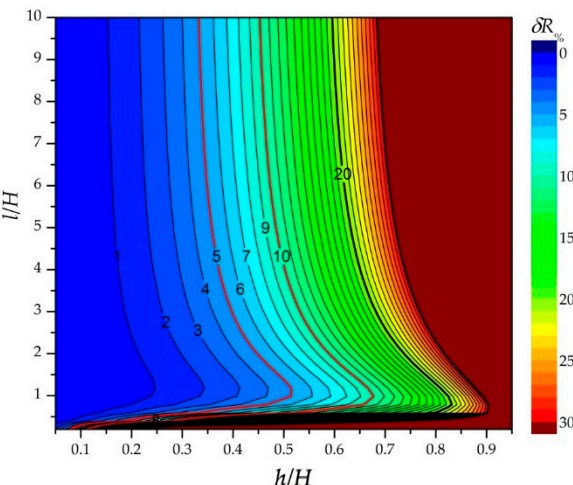

**Figure 12.** Percentage deviation between the resistance $\widetilde{R}$ calculated using Formula (36) for the real defect length and the resistance $R$ calculated numerically.

Assuming a conventionally acceptable error of estimation of 10%, Figure 12 shows that Formula (36) gives correct values for defects with a length not exceeding half of the width of the path, when the path length is greater than its width. The second condition is almost always met for typical conductive paths, but the first significantly limits the scope of the formula. However, this range can be significantly expanded by correcting the defect length appropriately, depending on the path length. Therefore, the following dimensionless values were introduced:

$$r = \frac{R}{R_0} \quad \widetilde{r} = \frac{\widetilde{R}}{\widetilde{R}_0} \quad v = \frac{h}{H} \quad w = \frac{l}{H} \quad g = \frac{h'}{h} \tag{48}$$

where $h'$ is the length of the defect for which $\widetilde{r} = r$. The test results show that the size $g$ is a quite mildly changing function of the variables $v$, $w$ and can be well approximated by the bilinear function:

$$\widetilde{g}(v, w) = avw + bv + cw + d \tag{49}$$

By minimizing the mean square approximation error,

$$\Delta g = \iint \left[ \widetilde{g}(v,w) - g(v,w) \right]^2 \mathrm{d}\,v\,\mathrm{d}\,w \tag{50}$$

The following values of $a$, $b$, $c$, $d$ coefficients were obtained by the standard approximation procedure:

$$a = -8.03E - 2 \qquad b = 6.70E - 2 \qquad c = -4.72E - 2 \qquad d = 0.807 \tag{51}$$

Taking into account Formulas (48), (49) it is finally possible to obtain an explicit formula for the corrected defect length:

$$\widetilde{h} = \left( a\frac{hl}{H^2} + b\frac{h}{H} + c\frac{l}{H} + d \right)h \tag{52}$$

The result of applying the corrected defect length in Formula (36) is shown in Figure 13. As in Figure 12, it presents the error in the calculated path resistance in relation to the numerically calculated resistance.

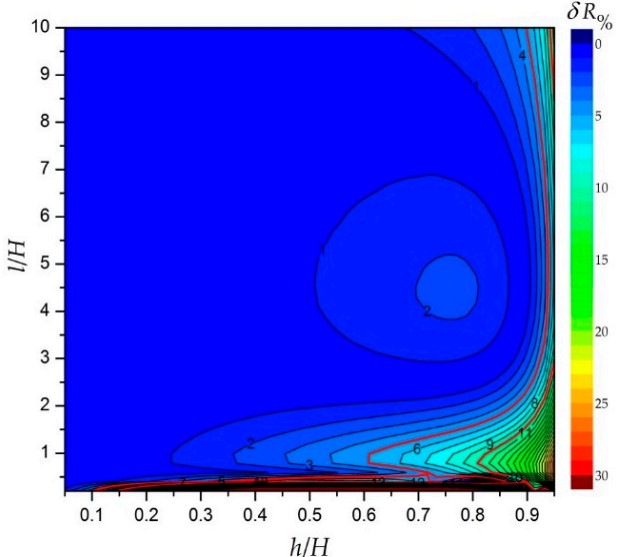

**Figure 13.** Percentage deviation between the resistance $\widetilde{R}$ calculated from Formula (36) for the corrected defect length according to Formula (51) and the resistance $R$ calculated numerically.

As can be seen, the error has been significantly reduced in the whole range of variables $h$ and $l$. Assuming, as before, an acceptable error of 10%, the range of applicability of Formula (36) was thus extended to defects reaching 95% of the path width.

## 7. The Results Comparision

Table 1 presents an example comparison of the results of calculations of layer resistance with an infinitely thin defect for the analytical model (formula (36)) without and with correction Formula (52), numerical simulations and measurements made in a physical model. The presented results are for the film which the length to width ratio is 4. It results from this comparison that calculations according to Formula (36) without taking into account the correction give acceptable results only for relatively small defects. The application of the correction Formula (52) significantly improves the accuracy of estimation and gives results very similar to numerical simulations and the experimental measurement results.

**Table 1.** The comparison of calculation results according to different models and measurements of the relative resistance $R/R_0$ of the layer with a defect depending on the relative length of the defect $h/H$.

| Relative Length of the Defect $h/H$ | Analytical Calculations According to the Formula (36) | Analytical Calculations According to the Formula (36) with Correction Formula (52) | Numerical Calculations | Measurements |
|---|---|---|---|---|
| 0.1 | 1.005 | 1.002 | 1.002 | $1.005 \pm 0.020$ |
| 0.2 | 1.022 | 1.010 | 1.008 | $1.012 \pm 0.020$ |
| 0.3 | 1.051 | 1.022 | 1.019 | $1.019 \pm 0.020$ |
| 0.4 | 1.097 | 1.040 | 1.036 | $1.039 \pm 0.021$ |
| 0.5 | 1.164 | 1.067 | 1.059 | $1.070 \pm 0.021$ |
| 0.6 | 1.264 | 1.104 | 1.090 | $1.097 \pm 0.022$ |
| 0.7 | 1.422 | 1.154 | 1.135 | $1.152 \pm 0.023$ |
| 0.8 | 1.700 | 1.224 | 1.205 | $1.205 \pm 0.024$ |
| 0.9 | 2.352 | 1.321 | 1.339 | $1.341 \pm 0.026$ |

## 8. Summary and Conclusions

In this paper, two models were presented enabling assessment of the impact of defects on the flow field distribution and the resistance of thin conductive layers. The defect in the analytical model has the shape of an ellipse arbitrarily oriented relative to the electric field, forcing the flow of current. Accurate formulas for the distribution and intensity of electric potential in the analyzed model were given, as well as formulas for estimating the impact of a defect on the conductive path resistance. The numerical model applies to a path of finite dimensions with an infinitely thin defect oriented perpendicular to the axis of the path. Boundary conditions at the path edges are included in the numerical method, whereas in the analytical model they are not. The algorithm and the program for the numerical procedure were created based on the boundary method of integral equations. The results derived from the numerical and analytical models were compared with measurements of a physical model. Satisfactory compliance was observed between the measurement results and the calculations from the numerical model in the whole range of system parameters in which variability was found, as well as with the calculations for the analytical model when the path length was greater than its width and the defect length did not exceed half of the path width. Outside of this range, the error in the analytical estimation of path resistance with a defect exceeded 10% and increased rapidly with increasing defect length. This error was caused by the fact that the analytical model does not take into account the boundary conditions at the path edges. However, it can be significantly reduced for defects with lengths exceeding up to 90% of the path width, significantly expanding the scope of application of the analytical formula. This can be achieved by substituting an appropriately corrected defect length in accordance with the Formula (52), which was obtained by means of mean-square approximation of the solution for a numerical model by a bilinear function. This improvement allows for swift and fairly accurate estimation of the impact of a defect on the conductive path resistance, without the need for numerical simulations.

To conclude, it should be stated that the results obtained based on the numerical model are the best consistent with the experiment. However, analytical approximation gives results very similar in almost the whole range of parameter variability of the analyzed system after taking into account the correction (51). It allows to quickly and accurately estimate the effect of a defect on the conductive layer resistance, without the need for numerical simulations

**Author Contributions:** Conceptualization, S.P. and E.K.; Data curation, S.P., J.P. and E.K.; Formal analysis, S.P. and J.P.; Investigation, S.P., J.P. and E.K.; Methodology, S.P., J.P. and E.K.; Supervision, S.P. and E.K.; Validation, S.P. and J.P.; Visualization, S.P. and J.P.; Writing—original draft, S.P., J.P. and E.K.; Writing—review & editing, E.K. All authors have read and agreed to the published version of the manuscript.

**Funding:** This research received no external funding.

**Conflicts of Interest:** The authors declare no conflict of interest.

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
