# Peer review of "Field Modeling the Impact of Cracks on the Electroconductivity of Thin-Film Textronic Structures"

_electronics, doi:10.3390/electronics9030402_

Round 1

Reviewer 1 Report

This is an interesting paper that deals with a problem that is gaining more attention on last decade and specialy on recent years. Although photonic sensing is going to replace mostly all electronic sensors including for wearable tex the work herein report is important and maybe give a positive contribution to the development of current and new wearable sensors of a wide variety of types. Theoretically the work is sound and we'll designed. The experimental work and the conclusions reached are also fine and sufficiently justified. I would just like to point out that further consideration must be taken on heat effect. It's positive and correct to use the same current for all measurements but the Joule effect must be further taken into account repeating the measurements with a range set of currents in the range of foreseeable current fluctuation in practical use but also driving it into limit or critical situations. Further simulation work can be easily added to the paper and probably at least preliminar measures can also be included in a revised (slightly) version of the paper to be accepted for publication.

Author Response

Dear Reviewer,

We would like to thank you very much for your time you spent on correcting our paper and valuable suggestions which helped us improve our paper.

Below we have written answers for your comments. All of them are placed directly under the comments to make it easier to find our answers. Additionally, all changes in the paper were highlighted with yellow:

Reviewer #1 (Technical Comments to the Author):

This is an interesting paper that deals with a problem that is gaining more attention on last decade and specialy on recent years. Although photonic sensing is going to replace mostly all electronic sensors including for wearable tex the work herein report is important and maybe give a positive contribution to the development of current and new wearable sensors of a wide variety of types. Theoretically the work is sound and we'll designed. The experimental work and the conclusions reached are also fine and sufficiently justified. I would just like to point out that further consideration must be taken on heat effect. It's positive and correct to use the same current for all measurements but the Joule effect must be further taken into account repeating the measurements with a range set of currents in the range of foreseeable current fluctuation in practical use but also driving it into limit or critical situations. Further simulation work can be easily added to the paper and probably at least preliminar measures can also be included in a revised (slightly) version of the paper to be accepted for publication.

We would like to thank the reviewers for the important suggestions. We fully agree with the reviewer's opinion that it would be advisable to extend the research described in the article, including thermal effects. Performing such experiments using the existing measurement system is possible, but developing a numerical program for their exact simulation would be much more difficult.

As it is known, temperature affects the conductivity of the conductor, which indirectly affects the distribution of current density, which depends on the amount of generated Joule heat. It means that considering this effect on the current density distribution requires solving the problem of coupled fields - flow and temperature. It can be achieved by iterative field auto-negotiation, but such procedures are much more complex and do not always coincide. Besides, the formulation of the issue of calculating the temperature distribution in the analyzed physical system is difficult. They are associated with problems in determining boundary conditions adequate to the actual cooling conditions of the tested object, which from the air side takes place in various physical ways (conduction, convection, radiation). It should also be noted that such a problem would have to be formulated as a 3D problem, for example because of the different conditions of receiving heat on both sides of the textronic conductive layer.

Summing up, we consider the Reviewer's remark regarding the extension of research in the suggested direction to be very valuable, but in our opinion it is a very complex topic, requiring the development of a separate algorithm and numerical program. It can certainly be devoted to separate paper.

Attached please find the improved and upgraded text of our article.

Reviewer 2 Report

This paper describes analytical and numerical models for understanding phenomena related to the electrical conductivity of thin electroconductive textronic structures. Because of the unavoidable damage during use, it is necessary to study the defects of the conductive textile. The authors used algorithm in the numerical model to simulate the potential distribution and distribution of the current density vector, etc under defects. Simulations match experimental results. However, the manuscript has several issues, which need to be fully addressed, before it can be considered for publication in Electronics.

  1. Figure 2 shows the woven structure of the nylon textile. The mechanical and electrical properties of different textronic structures will be different, and the effect of defects on electrical properties will also be different. Other textile structures such as knitting are also recommended as experimental groups.
  2. The authors need to explain the advantages of their simulation methods over other methods in the paper. A table was recommended to be added in the manuscript.

The amended paper can be considered for publication.

Author Response

Dear Reviewer,

We would like to thank you very much for your time you spent on correcting our paper and valuable suggestions which helped us improve our paper.

Below we have written answers for your comments. All of them are placed directly under the comments to make it easier to find our answers. Additionally, all changes in the paper were highlighted with yellow:

Reviewer (Remarks to the Author):

This paper describes analytical and numerical models for understanding phenomena related to the electrical conductivity of thin electroconductive textronic structures. Because of the unavoidable damage during use, it is necessary to study the defects of the conductive textile. The authors used algorithm in the numerical model to simulate the potential distribution and distribution of the current density vector, etc under defects. Simulations match experimental results. However, the manuscript has several issues, which need to be fully addressed, before it can be considered for publication in Electronics.

Figure 2 shows the woven structure of the nylon textile. The mechanical and electrical properties of different textronic structures will be different, and the effect of defects on electrical properties will also be different. Other textile structures such as knitting are also recommended as experimental groups.

The structures with continuous electrical conductivity cannot be created on knitted fabrics without a surface leveling layer. The paper presents the results of work for an exemplary composite substrate, in which an additional polyurethane layer was applied to the fabric threads. Only such a fabric construction allows the production of thin electroconductive structures in the PVD process. Analysis regarding the comparison of the results of the created model for the resistance of thin electrically conductive layers created on other composite materials is planned at a later stage of research. Such information was placed in the paper.

The authors need to explain the advantages of their simulation methods over other methods in the paper. A table was recommended to be added in the manuscript.

The amended paper can be considered for publication.

We would like to thank the reviewer for the important suggestions. In response to the Reviewer's comments, section 4.2 has been supplemented. There the advantages of the used numerical simulation method are mentioned (first paragraph). Section 6 has been added. In this section a table allowing to compare the results of calculations for the analyzed models with the results of the measurements and the advantages of the simulation methods used in the summary were clearly emphasized (last paragraph).

Attached please find the improved and upgraded text of our article.